# Occurrence and Molecular Identification of Wild Yeasts from Jimma Zone, South West Ethiopia

**DOI:** 10.3390/microorganisms7120633

**Published:** 2019-11-30

**Authors:** Anbessa Dabassa Koricha, Da-Yong Han, Ketema Bacha, Feng-Yan Bai

**Affiliations:** 1State Key Laboratory of Mycology, Institute of Microbiology, Beijing 100101, China; korichaa@im.ac.cn (A.D.K.); ucashank@163.com (D.-Y.H.); 2College of Life Sciences, University of Chinese Academy of Science, Beijing 100049, China; 3Department of Biology, College of Natural Sciences, Jimma University, Jimma 378, Ethiopia

**Keywords:** South West Ethiopia, Yeast diversity, phyllospheres, rhizospheres, fruit, tree bark

## Abstract

Yeasts are common inhabitants of most fruit trees’ rhizospheres and phyllospheres. Wild yeasts are the major driving force behind several modern industrial biotechnologies. This study focused on determining the occurrence and frequency of wild yeasts associated with domestic and wild edible tree barks, fruits, and rhizosphere soil samples collected over two seasons (i.e., spring and summer) in South West Ethiopia. A total of 182 yeast strains were isolated from 120 samples. These strains belonged to 16 genera and 27 species as identified based on the sequence analysis of the D1/D2 domain of the large subunit (26S) ribosomal RNA gene. *Candida blattae, Pichia kudriavzevii, Candida glabrata, Saccharomyces cerevisiae*, and *Candida humilis* were the most dominant yeast species isolated from the bark samples. Only *Pichia kudriavzevii* was regularly detected from the bark, rhizosphere, fruit, and sugarcane samples. The retrieval of yeasts from bark samples was more frequent and diverse than that of soil, fruits, and sugarcane. The frequency of detection of yeasts during the spring was significantly higher than in the summer season. However, there was no significant seasonal variation in the frequency of detection of yeast species between the rhizosphere and phyllosphere samples.

## 1. Background

Plants generally associate with large numbers and diverse groups of microorganisms including yeasts, bacteria, filamentous fungi, viruses, protozoa, and algae. In the first place, plants are rich sources of nutrients for microorganisms [1]. The association between plants and microorganism could be competition, commensalism, mutualism, or parasitism [2]. Generally, the microbial communities of plant shoot and root systems play a pivotal role in plant functioning by inducing physiology and development [3]. Nowadays, plant–bacteria interactions are well studied, but yeast–plant interactions have attracted very little attention. Plant-associated microorganisms colonize different parts of a plant, where each plant part has its distinct microbiome [4,5]. Primarily, the rhizosphere habitats [5] and the endophytic microbiota [4] are among the most explored microbiomes.

The caulosphere (stems) is a wooden aerial plant part containing large groups of microorganisms [6]. Other than that, the phylloplane (leaves), anthosphere (flowers), and carposphere (fruits) are parts of the phyllosphere habitats required by numerous microorganisms for colonization and the acquisition of nutrition, growth, and/or survival. Interestingly, a substantial group of bacteria, yeasts, and filamentous fungi have been retrieved from the phyllosphere of several plant species [7]. However, due to the fact of plants’ physical barriers and several chemical factors, the growth and survival of non-beneficial microbes on the phyllosphere is inhibited [1]. However, many microorganisms naturally inhabit different niches by using their biological mechanisms of adaptation [8] to enhance their chances of survival in both the plant rhizosphere and phyllosphere. Therefore, by establishing their lifestyle through free-living, latent pathogen, or symbiosis and growth promotion, microorganisms can interact with their residing hosts.

Fruits are one part of the phyllosphere habitat for microorganisms [7]. Similar to sugar-rich environmental samples, fruits are a common food source for many yeast species [9]. Beyond this, fruits are susceptible to contamination as a result of fruit exposure to the environment [10]. Soil, dust, water, and poor handling practices at harvest or during postharvest processing are major sources of fruit contamination. However, the possession of a tough cover, waterproofness, and low pH values serve as a barrier to successful invasion (entry) of the fruit tissue by most spoilage and pathogenic microbes [11]. In addition, fruit skin contains diverse groups of microbes encompassing both bacteria and fungi.

The rhizosphere is a micro-ecological zone surrounding and influenced by plant root secretions and is supposed to be the most complex microbial habitat on Earth [12,13]. It is a place of specialized ecological niches, where complex microbial interactions are accomplished. Microorganisms of the rhizosphere encompass bacteria, fungi, and protozoans, characterized either as helpful or deleterious species. Representatives of many of these microbial groups are well investigated and reported [14].

Generally, yeasts are ubiquitous groups of fungi and are associated with a wide range of ecological niches. Therefore, wild yeasts of different species can live in very diverse habitats including the air and soil [13], tree bark and decayed soil [15,16], leaf surfaces [17], seawater [18], rivers and lakes [19], on/in warm-blooded animals [20], and on fruit surfaces [21]. These associations are based on an adaptation of the organisms to that habitat for their survival.

Finally, yeasts are non-photosynthetic organisms that have been involved, since ancient times, in a wide range of food fermentations and live as decomposers, plant and animal pathogens, and tools for biotechnological research. Traditional fermentation products (such as staple foods, beverages, milk, and condiments) have been part of the human diet for millions of years [22,23]. In the beginning, fermentation of food depended on spontaneous fermentation by wild or naturally occurring yeasts and bacteria. In the recent development of science and technology, wild yeasts play a vital role in modern industrial biotechnology for the production of citric acid [24], single-cell proteins, ethanol [25], and microbial oils [26,27]. Hence, edible and non-edible environmental samples are principally anticipated to be the sources for theses yeasts.

The wild fruiting tree species *Ficus sycomorus* (L.) (Moraceae) and *Ficus vasta* (Moraceae), from natural and secondary forests, were subjected to wild yeast isolation. These fig trees have had a longstanding historic relationship with human beings as is evidenced in the literature and Holy Bible. Moreover, the frequencies of isolation of *Saccharomyces cerevisiae* from the barks of these trees are supportive indicators for the possible isolation of a diverse group of yeast species from wild fruit trees. In the current study, *Candida blattae* was isolated for the first time from fig tree bark samples. Thus, there will be the chance to retrieve a novel yeast strain or strains for significant application in industry and agriculture. Wild yeasts have been domesticated and have benefitted humankind for millennia and are wide-ranging in their fundamental and industrial importance in scientific, food, medical, and agricultural disciplines. Hence, a comprehensive study of all ecological zones in Ethiopia could certainly unearth many more yeast species, potentially including novel strains.

## 2. Materials and methods

### 2.1. Study Area and Sampling

Samples of tree bark (Harbuu (*Ficus sycomorus* (L.)), Qilxuu (*Ficus vasta*), and Hadaamii (*Euphorbia candelabrum*)), rhizosphere soil, fruits (Lemon (*Citrus limon* (L.)), Mango (*Mangifera indica*), and Guava/Zaytunii (*Psidium guajava*)), and sugarcane (*Saccharum officinarum*) were collected from two localities: Belete-Gera Forest (approximately 50 km to the west of Jimma) [28] and Jimma Town District (Table 1). All samples were collected carefully using sterile self-sealing plastic bags, transported to the laboratory, and processed within 2–4 h after collection. Samples were collected during two distinct seasons: summer 2016 (June to August) and spring 2017 (April and May). A total of 120 samples were examined (Table 1).

### 2.2. Yeast Isolation and Identification

At least a three-fold sample volume of enrichment medium made of YPD (w/v, 2% glucose, 2% peptone, and 1% yeast extract) broth supplemented with 200 μg/ mL chloramphenicol and 1 mL 1 M HCl was added to the samples in order to ensure stringent conditions during incubation. The tubes were closed loosely and incubated at 25 °C. Gas formation, indicative of fermentation, was checked for every day for 14 days. After gas formation, 2 μL of the liquid was spread on an agar plate prepared using the YPD broth supplemented with 2% (*w*/*v*) agar. The plates were incubated at 25–28 °C for 3–7 days. Representatives of each colony type grown on the solid medium were purified and maintained for the next analysis.

Fruit samples were chopped into small pieces. Approximately 2–4 g samples, composed of skin and pulp, were placed in 100 mL Erlenmeyer flasks containing 10 mL of YPD broth, supplemented with 200 μg/mL of chloramphenicol, and shaken on a rotary shaker (100 r/min). Every 1 day, during incubation at 25 °C, 0.1 mL aliquots were surface spread onto YPD agar, and morphologically distinct colonies were picked after an appropriate incubation period, purified by repeated plating, and maintained on YPD agar at 4 °C.

### 2.3. Amplification of the D1/D2 Domains of the Large Subunit (LSU) rDNA Gene

The purified yeast isolates were activated on YPD media for 24 h at 30 °C, and the template DNA extraction carried out from the yeast colony by the method described by Makimura et al. [29]. The D1/D2 domain of the 26S rDNA gene amplified using the primers NL1 (5’-GCA TAT CAA TAA GCG GAG GAA AAG-3’) and NL4 (3’-GGT CCG TGT TTC AAG ACG G-5’). The PCR reaction was performed in a total reaction volume of 25 μl, and PCR amplification carried out under the following conditions: an initial denaturation at 95 °C for 5 min, followed by 35 cycles at 94 °C for 2 min, 52 °C for 1 min, 72 °C for 2 min, and a final extension at 72 °C for 10 min. The amplicons were sequenced using the method described by Bai et al. [30]. The obtained sequences of the 26S rDNA D1/D2 domains of the strains analyzed were searched against the GenBank database using the basic local alignment search tool (BLAST) (https://blast.ncbi.nlm.nih.gov/Blast.cgi) at the National Center for Biotechnology Information (NCBI) to find the top matches to the sequences compared. A threshold of 99% or above of sequence identity with the type of strains related to the described yeast species in the rDNA region were used for species identification [31,32]. The DNA sequences representing the species were deposited in the NCBI database (https://www.ncbi.nlm.nih.gov/) with the accession numbers (MN075224–MN075250) (Table 2).

### 2.4. The D1/D2 Phylogeny Construction

All the D1/D2 sequences of the yeast strains were aligned by CLUSTALW, along with reference sequences for the individual species previously deposited in GenBank. The representative phylogenetic tree of the aligned sequences was constructed by neighbor joining using MEGA software version 6 [33].

### 2.5. Habitat Preference of Yeast Isolates

The two-sample *t*-tests of unequal variances were used to examine the statistical significance in yeasts’ preference to inhabiting the phyllosphere over the rhizosphere or vice versa. The null hypothesis of the test was that the yeasts show no preference between the phyllosphere and rhizosphere.

### 2.6. Seasonality of Yeast Detection

The statistical significance of the mean difference in the prevalence of yeasts between summer and spring seasons was examined using two-sample *t*-tests of unequal variances. The null hypothesis of this test was that there was no significant difference in yeast species prevalence between summer and spring.

The statistical significance differences among means was analyzed using unequal variance *t*-tests by JMP^®^ 13 statistical software, JMP, SAS Institute Inc., Cary, NC, USA [34].

## 3. Results

### 3.1. Frequency of Isolation and Loads of Yeasts among Sample Sources

In this study, 182 yeast strains were isolated from 120 samples of tree barks, rhizosphere soil, fruits, and sugarcane. These strains belonged to 16 genera and 27 species were identified based on the sequence analysis of the 26S rDNA D1/D2 domain (Table 2, Figure 1). The yeast counts (CFU/g) varied from 1.30 × 10^2^ to 2.21 × 10^5^ (Table 1). The species *Candida blattae, Pichia kudriavzevii, Meyerozyma guilliermondii, Candida humilis*, and *Saccharomyces cerevisiae* were the most frequently isolated yeasts (Table 3). Whereas, *Pichia kudriavzevii* was the only yeast species associated with all sample sources (including barks, rhizosphere soil, sugarcane, and fruits) except for the mango and guava samples.

Among the strains, 99 (54.4%), 33 (18.13%), 45(24.72%), and 5 (2.75%) were isolated from barks, rhizosphere soil, fruits, and sugarcane, respectively (Table 3). The D1/D2 domain region of the large subunit of 26S rDNA was used to differentiate the strains to the species. A majority (26/27) of the identified species belonged to ascomycetous yeasts, encompassing a total of 176 (96.7%) of the isolated strains (Table 2). Basidiomycetous yeasts were represented by only one species consisting of 6 (3.3%) of the identified strains.

### 3.2. Comparison of Yeast Counts

Even though no research reports on of the microbiology of Harbuu, Qilxuu, and Hadaamii barks, we tried to demonstrate the total yeast colony count (mean ± standard deviation) of Harbuu bark was 1.15 × 10^5^ ± 2.9 × 10^4^ CFU/g; Qilxuu barks was 1.14 × 10^5^ ± 3.0 × 10^4^ CFU/g; and for Hadaamii barks 1.06 × 10^4^ ± 1.4 × 10^3^ CFU/g. As well, the yeast colony count (mean ± standard deviation) of rhizosphere was 1.29 × 10^5^ ± 4.2 × 10^4^ CFU/g. Similarly, for fruits, the yeast colony count (mean ± standard deviation) of lemon was 7.71 × 10^2^ ± 2.7 × 10^2^ CFU/g, mango 4.10 × 10^2^ ± 2.5 × 10^2^ CFU/g, and guava 4.25 × 10^2^ ± 2.5 × 10^2^ CFU/g. Lastly, the yeast colony count (mean ± standard deviation) of sugarcane was 3.86 × 10^2^ ± 1.94 × 10^2^ CFU/g (Table 1 and Table 3).

### 3.3. Species Distribution among Sample Sources

Of the 27 yeast species identified, *Pichia kudriavzevii* (common species) and *Meyerozyma guilliermondii* were widely distributed among the majority of the environmental samples. *Candida blattae* was detected from all tree barks and some fruits (Table 3). *Pichia kudriavzevii* was the next frequently isolated species, being absent only from the mango and guava samples.

*Saccharomyces cerevisiae,* the common species associated with oaktree bark, was retrieved from the barks of both Harbuu (*Ficus sycomorus* L. (Moraceae)) and Qilxuu (*Ficus vasta* (Moraceae)), and lemon fruits. Of the total yeasts isolated and identified, as many as 15 of the yeast species were isolated from only one type of environmental samples analyzed in the current study (specifically from *Ficus sycomorus* (L.) (Table 3).

Accordingly, among the environmental samples subjected to yeast isolation, Harbuu (*Ficus sycomorus* L. (Moraceae)), Qilxuu (*Ficus vasta* (Moraceae)), and lemon fruits were sources of higher numbers of yeast species, accounting for 15 (55.56%), 13 (48.15%), and 12 (44.44%) for different yeast species of all the species identified, respectively. In contrast, the bark of Hadaamii (*Euphorbia candelabrum*) and some fruit samples, namely, mango and guava, contributed a lower number of yeast species. Accordingly, the bark of Hadaamii and mango and guava fruits, respectively, contributed only 4 (14.81%), 5 (18.52%), and 5 (18.52%) species to the current pools of yeast species (Table 3).

The incidence of yeast species in different samples of the same environmental samples also varied. For instance, only one yeast species (*Pichia kudriavzevii*) was identified regularly from all the rhizosphere samples analyzed. Besides, a total of 15 yeast species were isolated from 40 samples of Harbuu barks, at the rate of one to two species per individual sample (1.52 species per sample on average). *Candida blattae*, the pioneer and most frequently isolated yeast species, was encountered in 16 (35.56%) of the 40 samples analyzed, followed by *Saccharomyces cerevisiae*, *Pichia kudriavzevii,* and *Candida glabrata*; each of them was found in four of the samples analyzed. Surprisingly, seven yeast species (*Aureobasidium pullulans Candida pararugosa, Hanseniaspora opuntiae*, and *Saccharomycopsis malanga*) were isolated from the only sample collected from the bark of Harbuu (*Ficus sycomorus* L. (Moraceae)).

Of the 13 species isolated from 40 samples of Qilxuu bark, *Candida blattae* was the most dominant yeast which was isolated from 16 (40.0%) of the sampled Qilxuu barks. *Pichia kudriavzevii* was the second dominant species which was found in 10 (25.0%) of the samples. From individual samples, only one to two species were retrieved. *Saccharomyces cerevisiae* and *Candida glabrata* were the third most frequently isolated species, each isolated from five (12.5%) of the collected samples.

Hadaamii bark was the sample from which the least number of yeast species were isolated. *Candida blattae, Candida catenulata, Candida humilis*, and *Pichia kudriavzevii* were species identified from Hadaamii bark samples. The frequency of detection of yeast strains per individual sample was one to two species.

Rhizosphere samples contributed a total of eight yeast species, isolated from seven samples. Also, each sample contained three to seven yeast species. The most frequently isolated yeast species was *Pichia kudriavzevii* and was encountered in all seven (100%) samples. *Meyerozyma guilliermondii*, the anamorphic form usually known as *Candida guilliermondii*, and *Kluyveromyces marxianus* were isolated from 5 (71.43%) and 3 (42.86%) samples, respectively. *Candida albicans, Candida humilis, Candida intermedia, Hanseniaspora uvarum,* and *Lachancea thermotolerans* were species isolated from the rhizosphere in a frequency of 6.1% each. Surprisingly, *Saccharomyces cerevisiae* was not found in all rhizosphere samples. The representative strains are indicated in Table 3.

Among the fruits, lemon harbored the highest diversity of yeast species. *Kodamaea ohmeri* was the dominant yeast species 5 (18.52%) isolated from lemon fruits (Table 3). A total of twelve yeast species were isolated from fifteen lemon samples, but each sample contained no more than one to three yeast species. Mango also harbored diverse groups of yeast, contributing five species and *Hanseniaspora uvarum* being the dominant species. Guava, or Zaytun, is another kind of fruit from which yeast species were isolated. Among the five yeast species isolated from guava, *Meyerozyma guilliermondii* and *Kodamaea ohmeri* had a relatively higher frequency of detection with *Meyerozyma guilliermondii* encountered in 3 (75%) of the total guava samples. Likewise, the sugarcane samples contained five different yeast species, the dominant ones being the species of *Geotrichum silvicola*, *Kluyveromyces marxianus*, *Meyerozyma guilliermondii*, and *Pichia kudriavzevii*, each of which was detected in one (20%) of the samples. Each sample contained only one yeast species.

### 3.4. Effect of Seasons and Sample Type Analysis

Seasons and sample sources appeared to have an impact on the number and type of yeast isolates. The prevalence of yeasts varied among the two seasons (Table 4). There were highly significant differences in the prevalence of yeasts between the spring (M = 5.54, SD = 6.01, *n* = 24) and the summer season (M = 2.45, SD = 2.56, *n* = 20); as *t*(32) = 2.28, *p* = 0.03. However, the diversity and frequency of yeasts (Table 3) in the phyllosphere (M = 5.52, SD = 7.11, *n* = 27) and the rhizosphere (M = 4.13, SD = 3.8, *n* = 8) was not significantly different, since *t*(23) = 0.73, *p* = 0.24.

## 4. Discussions

Wild edible trees have evolved with a plethora of yeasts having vital roles for plant growth and health. In this study, we isolated diverse groups of yeasts from both the rhizosphere and the phyllosphere of wild and domestic edible fruiting trees. Among the yeast isolates presented here, 26 species belonged to the phylum Ascomycota and which were much more abundant than those from the phylum Basidiomycota with only one species. In the recent decade a considerable recorded data was available on the diversity and dynamics of plant microbiota as well as their potential in biotechnological application for the isolated community members.

### 4.1. Gene Sequence Comparisons

The coding region of the D1/D2 variable domains of the large subunit (LSU) rDNA or the complete small subunit (SSU) and the non-coding internal transcribed spacer (ITS) region of the ribosomal DNA were primarily emphasized for the yeasts’ molecular systematics [14]. Currently, we used the D1/D2 variable domain of the LSU rDNA to differentiate the yeasts isolated from the different edible tree samples as the results of easy to PCR (universal primers), short sequenced (400–650 bp), simple to align, variable enough to distinguish most of the yeast species, and universally available in a database for all known yeast species and is also used earlier than the ITS region [31,32,35]. The ITS region, including the 5.8S rDNA gene (coding and conserved) and two flanking variable and non-coding regions ITS1 and ITS2, shows low intraspecific variability and high interspecific polymorphism [35]. Therefore, the primary DNA barcode marker for yeast identification is the D1/D2 domain, although the ITS region has been chosen as a universal DNA barcode marker for general fungi [36].

### 4.2. Yeast Community Compositions of Different Samples

Harbuu (*Ficus sycomorus* L.) is a wild fig tree which produces fruits palatable for human beings and other organisms including birds, goats, monkeys, baboons, and sheep [37]. The microbiology of Harbuu bark, figs, and rhizosphere soil has still not been reported thus far. But yeasts recovered from similar wild sources were involved in biotechnological research and industrial production of fuel ethanol, feeds, fodder, and industrial enzymes [14]. Kurtzman et al. [14] demonstrated the industrial potential of the genera *Pichia* and *Kluyveromyces* for the production of single-celled proteins, exopolysaccharides, and fine chemicals. Ethanol is produced by fermentation of sugars by yeasts. Rao et al. [16] isolated the ethanol-producing yeast genera *Pichia, Candida, Kluyveromyces, Issatchenkia, Zygosaccharomyces*, *Clavispora, Debaryomyces, Metschnikowia, Rhodotorula*, and *Cryptococcus* from fruits and other tree barks. Currently, the genera *Debaryomyces*, *Zygosaccharomyces*, *Clavispora*, *Metschnikowia*, and *Cryptococcus* were not isolated from Harbuu (Table 3), but the remaining four genera, including *Pichia*, *Candida*, *Kluyveromyces*, and *Rhodotorula*, were retrieved from Harbuu bark samples. Sarris and Papanikolaou [27] also indicated the excellent candidacy of *R. mucilaginosa*, isolated from the surface of marine fish, for industrial applications of single-cell oil production. *Candida blattae*, a sister taxon in the *Candida intermedia* clade, was a predominant yeast species isolated from Harbuu barks. But, reports by Nguyen et al. [38] point out that *Candida blattae* is associated with the gut of basidioma-feeding beetles. *Saccharomyces cerevisiae* is another yeast strain most frequently isolated from Harbuu barks. It is the model eukaryotic organism utilized for biotechnological research worldwide. A previous study by Rachamontree et al. [39] showed *Saccharomyces cerevisiae*, isolated from cassava west, had the potential for single-cell protein production.

Qilxuu (*Ficus vasta*) is another fig tree species known for its production of figs or edible fruits. Fresh and dried figs are consumed during normal times and when there is a shortage of foods. Like Harbuu (*Ficus sycomorus* L.), Qilxuu (*Ficus vasta*) fruits are also commonly consumed by human beings, birds, goats, monkeys, baboons, and sheep [37]. Currently, we isolated 49 yeast strains of 13 different species from 40 samples of Qilxuu barks. The *Saccharomyces cerevisiae*, *Yarrowia lipolytica*, and *Pichia kudriavzevii*, recognized for industrial production of single-cell proteins and microbial oils, were isolated from Qilxuu barks. Besides, *Yarrowia lipolytica* is reported for industrial production of erythritol and mannitol [40], feedstock bioprocesses [41], and recombinant protein [42]. The genera *Candida, Pichia*, and *Debaryomyces* were recovered from different environmental sources and are utilized in biomass hydrolysates and production of l- or d-arabitol [43]. Budding yeasts (*Saccharomyces cerevisiae*), usually associated with human-related fermentations [44,45] and oak trees and/or their associated substrates (including damaged fruit, tree bark, rotten wood, and soil) [46], have also been recovered from Qilxuu barks. Bashir and Kassim [47] determined the occurrence of *Aureobasidium pullulans* from fig leaf; however, we couldn’t isolate this species from Qilxuu barks. Besides that, no basidiomycetous yeast species were isolated from Qilxuu bark samples.

Hadaamii (*Euphorbia candelabrum*) is a succulent tree species, commonly known as a candelabra tree. Even though there are no recorded data for the occurrence and diversity of yeast species associated with Hadaamii, we isolated *Candida blattae*, *Candida catenulata*, *Candida humilis*, and *Pichia kudriavzevii* from five samples with a low frequency of isolation—an average rate of one yeast species per sample.

The rhizosphere is a narrow region of soil influenced by root exudates and where strong microbial–plant interaction takes place [48]. Yeasts are among the most common microbial constituents of rhizosphere soil [49]. Vadkertiová et al. [50] reported the occurrences of 32 ascomycetous and 27 basidiomycetous yeast species from soil adjacent to the Rosaceae family. In this study, a total of 33 yeast strains from eight different species were isolated from rhizosphere soil. Similarly, Sarabia et al. [49] listed seven yeast species belonging to six genera from the rhizosphere, where *Meyerozyma guilliermondii* and *Candida railenensis* were the most frequently isolated species. However, *Candida railenensis* was not detected in the current study. *Pichia kudriavzevii* was the dominant yeast species (36.36%) of all the isolated species from the rhizosphere soil. Interestingly, the strain *Pichia kudriavzevii*, in addition to single-cell protein production, is reported to have the potential for ethanol and phytase production [39]. Furthermore, *Pichia kudriavzevii, Candida humilis,* and *Lachancea thermotolerans* isolated from both bark and rhizosphere soils can be considered as resident yeast species. The yeast diversity of the rhizosphere soil depends on the season, type and depth of the soil, plant species, and the locality [50,51]. The current results also support the observation that there was a significant difference (*p* < 0.05) in the prevalence of yeast species collected during the two seasons (i.e., spring and summer). But there were no significant differences between the phyllosphere and the rhizosphere samples.

Fruits are an important habitat for different groups of wild yeasts. Fruit microbiology has already been addressed by several authors from different countries [16,35,52,53]. Buenrostro-Figueroa et al. [54] reported that the species *Candida kefir, Kluyveromyces bulgaricus, Kluyveromyces fragilis, Candida macedoniensis*, and *Candida tropicalis* have been isolated from mango fruit. Currently, we retrieved yeast species including *Debaryomyces hansenii, Debaryomyces prosopidis, Aureobasidium pullulans, Hanseniaspora uvarum, Lodderomyces elongisporus, Rhodotorula mucilaginosa*, and *Yarrowia lipolytica* from mango and other fruits. The industrial significance of these strains, *Debaryomyces* spp., *Rhodotorula mucilaginosa*, and *Yarrowia lipolytica*, for the production of single-cell oil, endo-polysaccharides, and polyols when they grow on biodiesel-derived glycerol (Gly) were determined in a previous work of by Filippousi et al. [55]. Besides that, *Yarrowia lipolytica* strains are indispensable in modern biotechnological industry for microbiological processes of citric acid [24,56], biomass rich in proteins, oils, organic acids, enzymes, and heterologous proteins productions [57]. Abranches et al. [58] isolated 28 ascomycetous yeast species from 70 samples of guava fruit dominated by species of *Kloeckeraa fricana*, *K. apis*, *Issatchenkia* sp., *Pichia kluyverai*, and *P. membranifaciens*. In this study, we isolated only five ascomycetous yeasts species from four guava fruit samples.

From the perennial grass of the family Poaceae, sugarcane (*Saccharum officinarum*) was cultivated in different parts of Ethiopia. Currently, we isolated five yeast species from five sugarcane samples including *Geotrichum silvicola, Kluyveromyces marxianus, Meyerozyma guilliermondii, Pichia kudriavzevii*, and *Yarrowia lipolytica*. Shehata [59] also reported 26 yeast species from 77 sugarcane samples. The author determined the most common species including *Saccharomyces carlsbergensis* var. alcoholophila, *Saccharomyces cerevisiae*, *Pichia membranaefaciens*, *Candida krusei*, *Torulopsis stellata*, *Candida guilliermondii*, *Pichia fermentans*, *Candida intermedia* var. ethanophila, and *Schizosaccharomyces prombe*. Similarly, Frisk et al. [60] and Limtong et al. [61] also reported 150 and 158 yeast strains isolated from 79 and 94 sugarcane leaf samples, respectively.

In general, different sample sources (i.e., tree barks, rhizosphere soil, fruits, and sugarcane) contributed different arrays of yeast species that varied in the frequency of their isolation and diversity depending on the season, locality, and type of source. Ethiopia, one of the six biodiversity-rich centers located in East Africa, could be a source of many more yeast species besides what we have reported in the current study. Hence, further study of all the agro-ecological zones of the country could certainly unearth many more yeast species, potentially including novel strains.

## 5. Conclusions

Wild and domestic edible fruiting trees are sources for diverse groups of yeast species. Yeasts isolated from tree barks have higher species diversity and frequency than fruits, rhizosphere soils, and sugarcane. *Candida blattae* was the dominant species identified and the strain was recovered for the first time from tree barks and fruit samples. The budding yeast *Saccharomyces cerevisiae* was isolated from Harbuu and Qilxuu barks at a higher frequency but was not detected in rhizosphere soil, sugarcane, mango, or guava samples. *Pichia kudriavzevii*, *Candida humilis*, *Hanseniaspora uvarum, Meyerozyma guilliermondii*, and *Lachancea thermotolerans* isolated from both bark and rhizosphere could be considered resident yeast species. Moreover, a greater diversity and frequency of yeast species were recovered during the spring season than in the summer season.

## Figures and Tables

**Figure 1 microorganisms-07-00633-f001:**
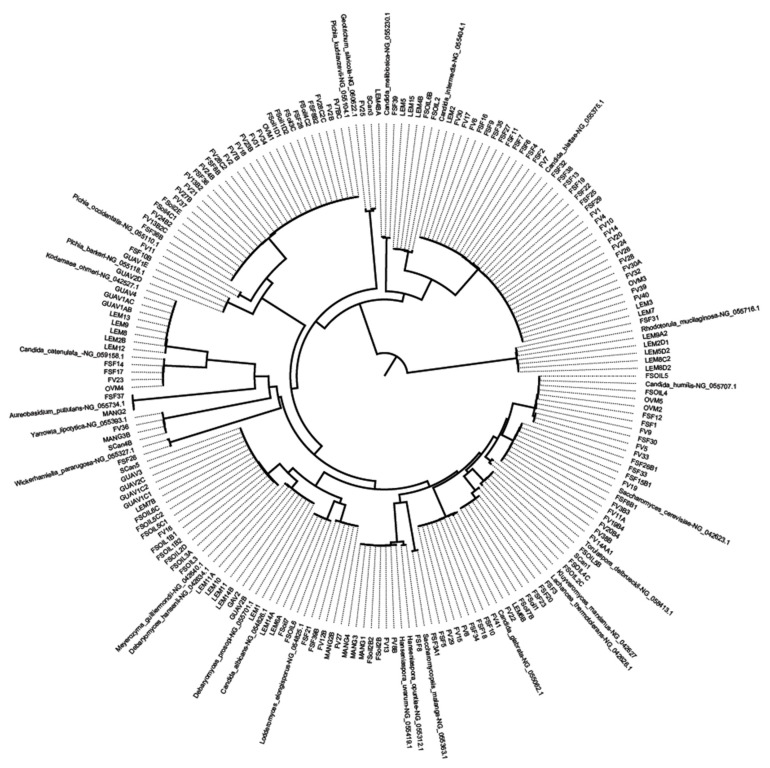
Phylogenetic tree depicting the phylogenetic relationships of the yeast strains isolated from tree bark, rhizosphere, and fruit samples with the type of the strains of the described yeast species concerned. The phylogram was constructed by neighbor-joining analysis based on the sequences of the 26S rDNA D1/D2 domain, and the reference sequences were retrieved from the NCBI database.

**Table 1 microorganisms-07-00633-t001:** The samples collected from the Jimma Zone, Oromia Regional State, southwestern Ethiopia and yeast counts in the samples.

Sample ID	Common Name	Scientific Name	Sample Type	Sampling Location	No. of Samples in		Ranges of Yeast Count (CFUs)
					Spring	Summer	
1	Harbuu/Shola	*Ficus sycomorus* (L.)	Bark	BF	20	20	8.00 × 10^4^–2.21 × 10^5^
2	Qilxuu/Warka	*Ficus vasta*	Bark	BF	20	20	7.60 × 10^4^–2.17× 10^5^
3	Hadaamii/Kulkual	*Euphorbia candelabrum*	Bark	BF	3	2	9.00 × 10^3^–1.30 × 10^4^
7	Rhizosphere soil		Soil	BF	4	3	6.50 × 10^4^–2.15 × 10^5^
4	Lemon	*Citrus Limon* (L.)	Fruit	JD	8	7	3.70 × 10^2^–1.08 × 10^3^
5	Mango	*Mangifera indica*	Fruit	JD	2	2	1.70 × 10^2^–8.10 × 10^2^
6	Guava/Zayitunaa	*Psidium guajava*	Fruit	JD	2	2	1.30 × 10^2^–7.40 × 10^2^
8	Sugarcane	*Saccharum officinarum*	Wax	JD	3	2	1.40 × 10^2^–6.50 × 10^2^

BF—Belete Forest; JD—Jimma Town District.

**Table 2 microorganisms-07-00633-t002:** **Yeast** species isolated from different environmental sample sources.

Sample ID	Species Name	Sample Sources	Strain ID	Accession Numbers	Sequence Identity (%) with Reference sp.
1	*Aureobasidium pullulans*	Mango fruit	M-2A	MN075224	100
2	*Candida albicans*	Lemon fruit	L-5	MN075225	100
3	*Candida blattae*	H-Bark	H-15	MN075226	99.6
4	*Candida catenulate*	H-Bark	Q-16	MN075227	100
5	*Candida humilis*	H *-Bark	H-11	MN075228	100
6	*Candida glabrata*	Q-Bark	H-6	MN075229	99
7	*Candida intermedia*	Rhizosphere	R-7	MN075230	99.4
8	*Candida melibiosica*	Lemon fruit	L-3	MN075231	100
9	*Candida pararugosa*	H-Bark	Q-45A	MN075232	100
10	*Debaryomyces hansenii*	Guava fruit	G-1	MN075234	100
11	*Debaryomyces prosopidis*	Lemon fruit	L-6	MN075233	99.1
12	*Geotrichum silvicola*	Sugarcane	Sc-3	MN075235	99.3
13	*Hanseniaspora opuntiae*	H-Bark	HD-4	MN075237	100
14	*Hanseniaspora uvarum*	Q-Bark	HD-6	MN075236	100
15	*Kluyveromyces marxianus*	Sugarcane	SC-1	MN075238	99.04
16	*Kodamaea ohmeri*	Guava fruit	G-1B	MN075239	100
17	*Lachancea thermotolerans*	Rhizosphere	R-11	MN075240	99.7
18	*Lodderomyces elongisporus*	Mango fruit	M-3	MN075241	99.2
19	*Meyerozyma guilliermondii*	Sugarcane	SC-4	MN075242	100
20	*Pichia barkeri*	Guava fruit	G-2B	MN075243	99.8
21	*Pichia occidentalis*	Q-Bark	H-1	MN075244	100
22	*Pichia kudriavzevii*	Rhizosphere	R-4	MN075245	100
23	*Rhodotorula mucilaginosa*	Mango fruit	M-5	MN075246	100
24	*Saccharomyces cerevisiae*	H-Bark	Q-44	MN075247	99.3
25	*Saccharomycopsis malanga*	H-Bark	Q-3	MN075248	99.6
26	*Torulaspora delbrueckii*	Q-Bark	H-12	MN075249	99.5
27	*Yarrowia lipolytica*	Sugarcane	SC-2	MN075250	99.2

Sample sources: H-bark—Harbuu bark; H *-Bark—Hadaamii bark; Q-Bark—Qilxuu bark.

**Table 3 microorganisms-07-00633-t003:** Number of strains and percentage of isolation of yeast species from tree barks, rhizosphere, and fruit samples collected from the study areas.

Species	*Ficus sycomorus* (L.)	*Ficus vasta*	*Euphorbia candelabrum*				Guava or	
Name	(Harbuu/Shola)	(Qilxuu/Warka)	(Hadaamii/Kulkual)	Rhizosphere	Lemon	Mango	Zeytun	Sugarcane
*Aureobasidium pullulans*	1 (2.22)					1(14.28)		
*Candida albicans*				2 (6.06)	2 (7.01)			
*Candida blattae*	16 (35.56)	16 (32.65)	1 (20)		3 (11.11)			
*Candida catenulata*	2 (4.44)	1 (2.04)	1 (20)					
*Candida glabrata*	4 (8.89)	5 (10.2)						
*Candida humilis*	3 (6.67)	3 (6.12)	2 (40)	2 (6.06)				
*Candida intermedia*				2 (6.06)	3 (11.11)			
*Candida melibiosica*	1 (2.22)				1 (3.7)			
*Candida pararugosa*	1 (2.22)							
*Debaryomyces hansenii*					3 (11.11)		1 (9.09)	
*Debaryomyces prosopidis*					2 (7.01)		1 (9.09)	
*Geotrichum silvicola*		1 (2.04)						1 (20)
*Hanseniaspora opuntiae*	1 (2.22)							
*Hanseniaspora uvarum*		2 (4.08)		2 (6.06)		3 (42.86)		
*Kluyveromyces marxianus*				3 (9.09)				1 (20)
*Kodamaea ohmeri*					5 (18.52)		3 (27.27)	
*Lachancea thermotolerans*	3 (6.67)			2 (6.06)	1 (3.7)			
*Lodderomyces elongisporus*	2 (4.44)	2 (4.08)				1 (14.28)		
*Meyerozyma guilliermondii*		1 (2.04)		8 (24.24)	1 (3.7)		4 (36.36)	1 (20)
*Pichia barkeri*	1 (2.22)						2 (18.18)	
*Pichia kudriavzevii*	4 (8.89)	10 (20.41)	1 (20)	12 (36.36)	1 (3.7)			1 (20)
*Pichia occidentalis*		1 (2.04)						
*Rhodotorula mucilaginosa*	1 (2.22)				4 (14.81)	1 (14.28)		
*Saccharomyces cerevisiae*	4 (8.89)	5 (10.2)			1 (3.7)			
*Saccharomycopsis malanga*	1 (2.22)							
*Torulaspora delbrueckii*		1 (2.04)						
*Yarrowia lipolytica*		1 (2.04)				1 (14.28)		1 (20)

**Table 4 microorganisms-07-00633-t004:** Lists of yeast species identified from tree bark, rhizosphere, fruit, and sugarcane samples collected during the spring and summer seasons.

Sample ID	Species Name	Spring	Summer	Total
1	*Aureobasidium pullulans*	2		2
2	*Candida albicans*	3	1	4
3	*Candida blattae*	26	10	36
4	*Candida catenulata*	4		4
5	*Candida glabrata*	7	2	9
6	*Candida humilis*	7	3	10
7	*Candida intermedia*	5		5
8	*Candida melibiosica*	2		2
9	*Candida pararugosa*		1	1
10	*Debaryomyces hansenii*	2	2	4
11	*Debaryomyces prosopidis*	2	1	3
12	*Geotrichum silvicola*	2		2
13	*Hanseniaspora opuntiae*	1		1
14	*Hanseniaspora uvarum*	4	3	7
15	*Kluyveromyces marxianus*	3	1	4
16	*Kodamaea ohmeri*	6	2	8
17	*Lachancea thermotolerans*	5	1	6
18	*Lodderomyces elongisporus*	3	2	5
19	*Meyerozyma guilliermondii*	11	4	15
20	*Pichia barkeri*	2	1	3
21	*Pichia kudriavzevii*	20	9	29
22	*Pichia occidentalis*		1	1
23	*Rhodotorula mucilaginosa*	4	2	6
24	*Saccharomyces cerevisiae*	9	1	10
25	*Saccharomycopsis malanga*	1		1
26	*Torulaspora delbrueckii*		1	1
27	*Yarrowia lipolytica*	2	1	3

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
