# Peer review of "Occurrence and Molecular Identification of Wild Yeasts from Jimma Zone, South West Ethiopia"

_microorganisms, 2019, doi:10.3390/microorganisms7120633_

Round 1

Reviewer 1 Report

Comments on manuscript 637863

The m/s by Dabassa et al, submitted for publication in the journal “Microorganisms”, presents potentially interesting information related with the occurrence and the molecular identification of several types of wild yeast species isolated from various edible tree barks, fruits and rhizosphere soil of several trees and plants. D1/D2 domain of the large subunit (26S) ribosomal RNA region was used to determine the species. Discussions and considerations in relation to the occurrence and the seasonal variability of the yeast strains isolated were performed.

The submitted m/s could present interest for the readers of “Microorganisms”. Nevertheless, a number of items need to be revised before acceptance of the paper.

1) Unlike the presentations done by the authors, in several types of fermentation processes that are very important for the modern Industrial Biotechnology (i.e. the ones of production of citric acid, single-cell oil, mannitol, arabitol, etc), utilization of wild-type microorganisms isolated from various natural habitats (i.e. soil, sourdoughs, marine fish, fruits, etc) has resulted in very interesting results that merit to be cited and discussed [see: Li et al. 2010 Biomass Bioenerg 34, 101–107; Liu et al. 2013 Marine Biotechnol 15, 26–36; Maina et al. 2017 Eng Life Sci 17, 333–344; Filippousi et al. 2019 J Appl Microbiol 127, 1080–1100]. In several of the above-mentioned papers, species that are presented in the current submission (i.e. Rhodotorula sp., Yarrowia lipolytica, Debaryomyces sp., etc) have been employed in the abave-mentioned bioprocesses.

2) For yeast species, identification the 5.8S-ITS rDNA region is considered to exhibit the highest resolving power for discriminating closely related fungal species (Schoch et al. 2012 Proc Natl Acad Sci USA 109, 6241–6246; Filippousi et al. 2019 J Appl Microbiol 127, 1080–1100). Please explain why this method was not used in the current submission.

3) There is no phylogenetic position of the newly yeast isolates and their comparison with reference-type wild yeast strains in a phylogenetic tree.

4) Much more discussions and considerations in relation to the biotechnological potential of several of the newly isolated yeast strains presented in the current submission (i.e. strains of Pichia sp. are capable to produce single-cell protein, strains of Yarrowia lipolytica are capable to produce citric acid and mannitol, strains of Debaryomyces sp. are capable to produce arabitol, strains of Saccharomyces sp. are capable to produce bioethanol; all these microorganisms are capable to grow in various waste- and residue-streams) are requested [see and comprehensively discuss: Liu et al. 2013 Marine Biotechnol 15, 26–36; RywiÅ„ska et al. 2013 Biomass Bioenerg 48, 148–166; Sarris et al. 2016 Eng Life Sci 16, 307–329; Filippousi et al. 2019 J Appl Microbiol 127, 1080–1100; Hoang Do et al. 2019 Microorganisms 7, 229].

Major revision in the points presented by the referee is requested.

Author Response

Responses to the specific comments from Reviewer #1.

Unlike the presentations done by the authors, in several types of fermentation processes that are very important for the modern Industrial Biotechnology (i.e. the ones of production of citric acid, single-cell oil, mannitol, arabitol, etc), utilization of wild-type microorganisms isolated from various natural habitats (i.e. soil, sourdoughs, marine fish, fruits, etc) has resulted in very interesting results that merit to be cited and discussed [see: Li et al. 2010 Biomass Bioenerg 34, 101–107; Liu et al. 2013 Marine Biotechnol 15, 26–36; Maina et al. 2017 Eng Life Sci 17, 333–344; Filippousi et al. 2019 J Appl Microbiol 127, 1080–1100]. In several of the above-mentioned papers, species that are presented in the current submission (i.e. Rhodotorula sp., Yarrowia lipolytica, Debaryomyces sp., etc) have been employed in the above-mentioned bioprocesses.

>> The importance of the newly isolated strain in modern Industrial Biotechnology for production of Microbial oil, Biomass, citric acid, single-cell oil, mannitol, and arabitol were incorporated in abstract, background, and discussion part and 17 additional references (14, 26, 48, 49, 50,51, 52, 53,54,55, 56, 57, 58, 59, 60, 61and 62) were cited.

For yeast species, identification of the 5.8S-ITS rDNA region is considered to exhibit the highest resolving power for discriminating closely related fungal species (Schoch et al. 2012 Proc Natl Acad Sci USA 109, 6241–6246; Filippousi et al. 2019 J Appl Microbiol 127, 1080–1100). Please explain why this method was not used in the current submission.

>> The Coding region of D1/D2 variable domains of the large subunit (LSU) of 26S rDNA or the complete small subunit (SSU) and the non-coding internal transcribed spacers regions of the ITS1 and ITS2 ribosomal DNA were majorly emphasized for yeasts molecular systematics (14).  Currently, we used the D1/D2 variable domain of the LSU of 26S rDNA to differentiate yeasts isolated from different edible tree samples. As the results of, easy to PCR (universal primers), Short sequence (400-650 bp), and simple for alignment, variable enough to distinguish most of the yeast species, and universally available database for all known yeast species. Nevertheless, the 26S rDNA D1/D2 domain was used for yeast identification [54, 55] earlier than the ITS region [60]. Therefore, the primary DNA barcode marker for yeast identification is the D1/D2 domain, although the ITS region has been chosen as a universal DNA barcode marker for general fungi [62].

There is no phylogenetic position of the newly yeast isolates and their comparison with reference-type wild yeast strains in a phylogenetic tree.

>> We incorporated the phylogenetic position of the newly yeast isolates and their comparison with reference-type of the wild yeast strains in a phylogenetic tree (Figure 1), on page 6.

Much more discussions and considerations in relation to the biotechnological potential of several of the newly isolated yeast strains presented in the current submission (i.e. strains of Pichia sp. are capable to produce single-cell protein, strains of Yarrowia lipolytica are capable to produce citric acid and mannitol, strains of Debaryomyces sp. are capable to produce arabitol, strains of Saccharomyces sp. are capable to produce bioethanol; all these microorganisms are capable to grow in various waste- and residue-streams) are requested [see and comprehensively discuss: Liu et al. 2013 Marine Biotechnol 15, 26–36; RywiÅ„ska et al. 2013 Biomass Bioenerg 48, 148–166; Sarris et al. 2016 Eng Life Sci 16, 307–329; Filippousi et al. 2019 J Appl Microbiol 127, 1080–1100; Hoang Do et al. 2019 Microorganisms 7, 229].

>> The industrial application potential of the new strains was incorporated in the abstract, Background and Discussion part with 17 additional references.

Reviewer 2 Report

In this manuscript, the authors isolated many yeast strains from tree barks, fruits, and rhizosphere soil in Ethiopia and reported the occurrence and frequency of yeast species by sorts of samples and seasons. They found that Pichia kudriavzebii strains were isolated from all samples and other several species were dominant in bark samples, rhizosphere and fruits. This study can be evaluated as a report on distribution of yeast species in natural samples from Ethiopia, however, the description and interpretation of results are partially inconsistent with the data and several required data and evidence are not provided.

Major comments:

In Abstract, C. blattae, C. humilis, P. kudriavzevii, and S. cerevisiae were listed as dominant species isolated from bark samples. But according to Table 3, the number of isolates of C. glablata was larger than that of C. humilis. kudriavzebii, C. humilis, and L. thermotolerans were listed as resident species both in bark and rhizosphere samples. Such description was not found in results and discussion sections. Furthermore, according to Table 3, L. thermotolerans was not isolated from rhizosphere and H. uvavum and M. guilliermondii were isolate from both bark and rhizosphere samples.

Thus, description and interpretation of results are inconsistent with the data. The authors should carefully reconsider the results.

For example, the following sentences are not consistent with Table 3.

"The species Candida blattae, Candida humilis, Kodamaea ohmeri, Pichia kudriavzevii, and Saccharomyces cerevisiae were the most frequently isolated yeasts."

"Candida blattae, the most frequently isolated yeast species, were encountered in 16 (35.56 %) of the 40 samples analyzed, followed by Saccharomyces cerevisiae, and Candida glabrata which were found in three and four of the samples, respectively."

The method for determination of D1/D2 domain should be described in detail. Which sample did the author use genomic DNA or cell suspension as the template for PCR? The sequences of the primers NL1 and NL4 should be listed or cite the appropriate reference.

How many samples did the author collect in each season? The number of samples should be divided into spring and summer in Table 1. In Table 4 and section 3.3, it is meaningless to compare the number of isolates between spring and summer without information of the number of samples collected in spring and summer.

In section 3.1, the yeast counts (CFU/g) were mentioned. These data are very important to evaluate the frequency of yeast species in each sample. Therefore, the range of yeast counts of each sample should be provided in Table 1, furthermore the yeast counts and the number of isolated colonies should be provided for all 120 samples as the supplementary information. Without these data, the frequency could not be argued.

  The order of Table 2 and 3 should be replaced because Table 2 shows the representative strains of each species and Table 3 shows overall results of this study.

Other specific comments

P1L19: "Samples were collected…." "were" should be added. The place where the samples were collected should be described. P3L7, 11, 13: Insert space before ul, g, r/min, respectively. P3L16: The 26S rDNA…."S" should be capitalized. Tables 1, 2, & 4: S/n should be defined. Limon should be Lemon in Table 1 & 2. P5L1: Insert space before ul. P5, section 3.1. L6: "were" should not be in italics. P5, section 3.1. L12: "subunit of 26S rDNA" 26S should be added. P5, section 3.1. L14: Add species name. P5, section 3.1. L15-17: The sentence describing accession numbers should be deleted. P5, section 3.2. L5: Show the basis for "dominant species". P10 and other pages: Species name should be italicized. Please check carefully. P10, section 3.3.: M and SD should be defined. P11, section 4.1.: This section should be moved to Results.

14. P11L18: Delete human beings.

Author Response

Responses to the specific comments from Reviewer #2.

In Abstract, blattae, C. humilis, P. kudriavzevii, and S. cerevisiae were listed as dominant species isolated from bark samples. But according to Table 3, the number of isolates of C. glablata was larger than that of C. humilis. C. kudriavzebii, C. humilis, and L. thermotolerans were listed as resident species both in bark and rhizosphere samples. Such description was not found in results and discussion sections. Furthermore, according to Table 3, L. thermotolerans was not isolated from rhizosphere and H. uvavum and M. guilliermondii were isolated from both bark and rhizosphere samples.

>> Yes, we corrected in this form, Candida blattae, Pichia kudriavzevii, Candida glabrata, Saccharomyces cerevisiae, and Candida humilis were the most dominant yeast species isolated from bark samples was corrected according to their abundance in the abstract.

>> Pichia kudriavzevii, Candida humilis, Hanseniaspora uvarum, Meyerozyma guilliermondii and Lachancea thermotolerans isolated from both bark and rhizosphere could be considered as resident yeast species was corrected in both abstract and discussed in the discussion part.

>> L. thermotolerans is isolated from both Bark and rhizosphere. This is correct.

The species Candida blattae, Candida humilis, Kodamaea ohmeri, Pichia kudriavzevii, and Saccharomyces cerevisiae were the most frequently isolated yeasts.

>> This was corrected to, Candida blattae, Pichia kudriavzevii, Meyerozyma guilliermondii, Candida humilis, and Saccharomyces cerevisiae were the most frequently isolated yeasts.

Candida blattae, the most frequently isolated yeast species, were encountered in 16 (35.56 %) of the 40 samples analyzed, followed by Saccharomyces cerevisiae, and Candida glabrata which were found in three and four of the samples, respectively

>> Candida blattae, the most frequently isolated yeast species, were encountered in 16 (35.56 %) of the 40 samples analyzed, followed by Saccharomyces cerevisiae, Pichia kudriavzevii, and Candida glabrata each of them were found in four of the samples analyzed.

The method for determination of D1/D2 domain should be described in detail. Which sample did the author use genomic DNA or cell suspension as the template for PCR? The sequences of the primers NL1 and NL4 should be listed or cite the appropriate reference.

>> We corrected under the title, 2.3. Amplification of the D1/D2 domains of the large subunit (LSU) rDNA gene

The purified yeast isolates were activated on YPD media for 24 h at 30°C and the template DNA extraction was carried out from the yeast colony by the method described by Makimura et al. [51]. The D1/D2 domain of the 26S rDNA gene was amplified using the primers NL1 (5-GCA TAT CAA TAA GCG GAG GAA AAG-3) and NL4 (5-GGT CCG TGT TTC AAG ACG G-3). The PCR reaction was performed in a total reaction volume of 25 μl and the PCR amplification was carried out under the following conditions: an initial denaturation at 95°C for 5 min, followed by 36 cycles at 94°C for 2 min, 52°C for 1 min, 72°C for 2 min; and a final extension at 72°C for 10 min. The amplicons were sequenced using the method as described in Bai et al. [52]. The obtained sequences of the 26S rDNA D1/D2 domains of the strains analyzed were searched against the GenBank database using the basic local alignment search tool (BLAST) at the National Center for Biotechnology Information (NCBI) (http://www.ncbi.nlm.nih.gov/BLAST/) to find top matches to the sequences compared. A threshold of 99% or above sequence identity with the type strains of related described yeast species in the rDNA region were used for species identification Kurtzman & Robnett [53] and Fell et al., [54].

How many samples did the author collect in each season? The number of samples should be divided into spring and summer in Table 1. In Table 4 and section 3.3, it is meaningless to compare the number of isolates between spring and summer without information of the number of samples collected in spring and summer.

>> The numbers of samples collected during both seasons (spring and summer) were listed in (Table 1).

In section 3.1, the yeast counts (CFU/g) were mentioned. These data are very important to evaluate the frequency of yeast species in each sample. Therefore, the range of yeast counts of each sample should be provided in Table 1, furthermore the yeast counts and the number of isolated colonies should be provided for all 120 samples as the supplementary information. Without these data, the frequency could not be argued.

>>The ranges of yeast counts (CFU/g) of each sample were presented in (Table 1) and the raw data for the colony number of 120 samples attached with the revised manuscript.

The order of Table 2 and 3 should be replaced because Table 2 shows the representative strains of each species and Table 3 shows overall results of this study.

>> The order of (Table 2) and (Table 3) were re-arranged according to the comment.

Other specific comments

P1L19: "Samples were collected…." "Were" should be added. The place where the samples were collected should be described. P3L7, 11, 13: Insert space before ul, g, r/min, respectively. P3L16: The 26S rDNA…."S" should be capitalized. Tables 1, 2, & 4: S/n should be defined. Limon should be Lemon in Table 1 & 2. P5L1: Insert space before ul. P5, section 3.1. L6: "were" should not be in italics. P5, section 3.1. L12: "subunit of 26S rDNA" 26S should be added. P5, section 3.1. L14: Add species name. P5, section 3.1. L15-17: The sentence describing accession numbers should be deleted. P5, section 3.2. L5: Show the basis for "dominant species". P10 and other pages: Species name should be italicized. Please check carefully. P10, section 3.3.: M and SD should be defined. P11, section 4.1.: This section should be moved to Results.14. P11L18: Delete human beings.

>> All these specific comments were incorporated in each line in line.

Round 2

Reviewer 1 Report

I have appreciated the work done in the revision of the submitted m/s. Though, a number of items of relatively minor nature remain to be clarified again in a second round of revision before acceptance of the paper:

1) In lines 27-39 (subchapter "2.3. Amplification of the D1/D2 domains of the large subunit (LSU) rDNA gene") there is some problem with the bibliographic references; I.e., it is stated that ... Makimura et al [51]..., while this ref. is the ref. [52]. Also it is indicated that ... Bai et al [52], while this is the ref. [53]. Please pay attention.

2) I have not seen the ref. previously suggested: Hoang Do et al. (2019) Microorganisms 7, 229. Under the same optics, please also cite the m/s by Vandermies and Fickers (2019) Microorganisms 7, 40, that had not been previously suggested. These papers should be cited and discussed since the current m/s is submitted in "Microorganisms".

3) I have seen that a new species namely "Debaryomyces prosopidis" has appeared. In order to discriminate Debaryomyces prosopidis from other Debaryomyces spp., besides other issues, we need also physiological tests (growth on sucrose or glucose solution at 30% w/v, if I remember well). Do you have information about this?

4) In a recent m/s, Filippousi et al (2019) presented significant growth and exceptional arabitol production by a completely new isolate of Debaryomyces sp.. Identification method had been based on 5.8S-ITS rDNA region. Please do correct and solid discussion.

5) Biomass and single-cell protein are almost the same. You could omit the terms and insted you could put only the term "single-cell biomass".

6) Oil produced by microorganisms (microbial oil) and single-cell oil are exactly the same terms. Please pay attection to the relevant parts of the "Introduction" and "Discussion" and use the correct terms.

Moderate revision in the points presented by the referee is presented.

Author Response

Responses to the specific comments from Reviewer #1. In lines 27-39 (subchapter "2.3. Amplification of the D1/D2 domains of the large subunit (LSU) rDNA gene") there is some problem with the bibliographic references; I.e., it is stated that ... Makimura et al [51]..., while this ref. is the ref. [52]. Also it is indicated that ... Bai et al [52], while this is the ref. [53]. Please pay attention.

>> The reference is corrected according to the comment.

The purified yeast isolates were activated on YPD media for 24 h at 30°C and the template DNA extraction was carried out from the yeast colony by the method described by Makimura et al. [52]. The D1/D2 domain of the 26S rDNA gene was amplified using the primers NL1 (5’-GCA TAT CAA TAA GCG GAG GAA AAG-3’) and NL4 (3’-GGT CCG TGT TTC AAG ACG G-5’). The PCR reaction was performed in a total reaction volume of 25 μl and the PCR amplification was carried out under the following conditions: an initial denaturation at 95°C for 5 min, followed by 36 cycles at 94°C for 2 min, 52°C for 1 min, 72°C for 2 min; and a final extension at 72°C for 10 min. The amplicons were sequenced using the method as described in Bai et al. [53]. The obtained sequences of the 26S rDNA D1/D2 domains of the strains analyzed were searched against the GenBank database using the basic local alignment search tool (BLAST) at the National Center for Biotechnology Information (NCBI) (http://www.ncbi.nlm.nih.gov/BLAST/) to find top matches to the sequences compared. A threshold of 99% or above sequence identity with the type strains of related described yeast species in the rDNA region were used for species identification Kurtzman & Robnett [54] and Fell et al., [55]. DNA sequences representing the species were deposited in the NCBI database (https://www.ncbi.nlm.nih.gov/) with the accession numbers (MN075224- MN075250) (Table 3).

I have not seen the ref. previously suggested: Hoang Do et al. (2019) Microorganisms 7, 229. Under the same optics, please also cite the m/s by Vandermies and Fickers (2019) Microorganisms 7, 40, that had not been previously suggested. These papers should be cited and discussed since the current m/s is submitted in "Microorganisms".

>> Both references were incorporated [63, 64]

3). I have seen that a new species namely "Debaryomyces prosopidis" has appeared. In order to discriminate Debaryomyces prosopidis from other Debaryomyces spp., besides other issues, we need also physiological tests (growth on sucrose or glucose solution at 30% w/v, if I remember well). Do you have information about this?

>>We didn’t do the physiological test the strain we isolated.

4). In a recent m/s, Filippousi et al (2019) presented significant growth and exceptional arabitol production by a completely new isolate of Debaryomyces sp. Identification method had been based on 5.8S-ITS rDNA region. Please do correct and solid discussion.

>> The 26S rDNA D1/D2 domain region is sufficiently variable to allow reliable identification of yeast species [54, 55, 65] and also used earlier than the ITS region [60]. The ITS region, including the 5.8S rDNA gene (coding and conserved) and two flanking variable and non-coding regions ITS1 and ITS2, show low intraspecific variability, and high interspecific polymorphism [65]. Therefore, the primary DNA barcode marker for yeast identification is the D1/D2 domain, although the ITS region has been chosen as a universal DNA barcode marker for general fungi [62].

5). Biomass and single-cell protein are almost the same. You could omit the terms and instead you could put only the term "single-cell biomass".

>> I omitted the rest and I used single-cell protein.

6). Oil produced by microorganisms (microbial oil) and single-cell oil are exactly the same terms. Please pay attention to the relevant parts of the "Introduction" and "Discussion" and use the correct terms

>> I used only single- cell oil.

Reviewer 2 Report

The manuscript has been revised well but there are still many minor issues which should be addressed before publication. Please consider all comments carefully.

P1.L13-14: Do not use capital letters for citric acid, single cell oil, biomass, single cell protein, ethanol, and microbial oils.

P1.L32: "associated" should be "associate".

P3.L8: Add "(Supplementary Table 1)" in the end of the sentence.

P3.L14: Insert space before "µl".

P3.L18: Insert space before "g".

P3.L29-30: Prime should be added after "5" and "3", e.g., "5-GCA" should be "5'-GCA".

P3.L39: The sentence describing accession numbers is necessary here.

"DNA sequences representing the species were deposited in the NCBI database (https://www.ncbi.nlm.nih.gov/) with the accession numbers (MN075224- MN075250) (Table 3)."

P4. Table 1: "Limon" should be "Lemon".

P5. "(Table 1)" should be added in the end of the sentence, "The yeast counts (CFU/g) varied from 1.30 x 102 to 2.21 x 105."

P5. Delete the sentence, "List of the isolated yeast species, sample sources, and accession numbers of the isolate’s gene sequences are as summarized in Tables 1 & 2."

P5. "(Table 1)" should be added in the end of the sentence, "The species Candida blattae, Pichia kudriavzevii, Meyerozyma guilliermondii, Candida humilis, and Saccharomyces cerevisiae were the most frequently isolated yeasts."

P5. "respectively (Table 3)" should be "respectively (Table 2)".

P5. "(Table 3)" should be added in the end of the sentence, "A majority (26/27) of the identified species belonged to ascomycetous yeasts encompassing a total of 176 (96.7%) of the isolated strains."

P6: Figure 1: Strain names were not defined in this manuscript and all sequence data were not deposited to the DNA database. If the author use this figure, the list of strain names with sample origin and accession numbers for 26S rDNA sequence should be provided as supplementary figures and all sequence should be deposited. Alternatively, phylogenetic three should be reconstructed with the sequences of 27 isolates in Table 3 and other type strains.

P6: "(Table 2 and Supplementary Table 1)" should be added in the end of the first sentence in section 3.2.

Table 2, 3, and 4: "Species Name" in Table 2, "S/n" in Table 3, and "ID" in Table 4 seem to indicate same numbers. Use one item. The order of D. hansenii and D. prosopidis and the order of H. opuntiae and H. uvarum in Table 2 are same as in Table 3 but different in Table 4. Check carefully.

Table 3: "Limon fruit" should be "Lemon fruit".

P9.L1,3,4,5: Lemon, Mango, an Guava should not be in italics.

P10L4: Do not use capital letter for rhizospere.,

P10L13: "(14)" should be "[14]".

P10L41: Do not use capital letter for single cell protein.

P11L9: Candida, Pichia, Debaryomyces should be in italics.

P11L30: Do not use capital letter for single cell protein.

P11L44: Debaryomyces should be in italics.

P11L45: Delete "(SCO)".

P11L47: Yallowia lypolytica should be in italics.

P11L48: Do not use capital letter for citric acid.

Author Response

Responses to the specific comments from Reviewer #2.   P6: Figure 1: Strain names were not defined in this manuscript and all sequence data were not deposited to the DNA database. If the author uses this figure, the list of strain names with sample origin and accession numbers for 26S rDNA sequence should be provided as supplementary figures and all sequence should be deposited. Alternatively, phylogenetic three should be reconstructed with the sequences of 27 isolates in Table 3 and other type strains.

>> The new phylogenetic tree was constructed by using the newly isolated strain (182) plus the 27 reference strains (Figure 1).

L13-14: Do not use capital letters for citric acid, single cell oil, biomass, single cell protein, ethanol, and microbial oils.

>>corrected

L32: "associated" should be "associate".

>>Corrected

L8: Add "(Supplementary Table 1)" in the end of the sentence.

>>Table 1 is referred.

L14: Insert space before "µl".

>>Space added.

L18: Insert space before "g".

>>Space added

L29-30: Prime should be added after "5" and "3", e.g., "5-GCA" should be "5'-GCA".

>>Indicated

L39: The sentence describing accession numbers is necessary here.

>>Incorporated

"DNA sequences representing the species were deposited in the NCBI database (https://www.ncbi.nlm.nih.gov/) with the accession numbers (MN075224- MN075250) (Table 3)."

>>Incorporated

Table 1: "Limon" should be "Lemon".

>>Changed to lemon

"(Table 1)" should be added in the end of the sentence, "The yeast counts (CFU/g) varied from 1.30 x 102to 2.21 x 105."

>>Table 1, is referred

Delete the sentence, "List of the isolated yeast species, sample sources, and accession numbers of the isolate’s gene sequences are as summarized in Tables 1 & 2."

>> Deleted

"(Table 1)" should be added in the end of the sentence, "The species Candida blattae, Pichia kudriavzevii, Meyerozyma guilliermondii, Candida humilis, and Saccharomyces cerevisiae were the most frequently isolated yeasts."

>>Table 1 is referred

"Respectively (Table 3)" should be "respectively (Table 2)".

>> Corrected

"(Table 3)" should be added in the end of the sentence, "A majority (26/27) of the identified species belonged to ascomycetous yeasts encompassing a total of 176 (96.7%) of the isolated strains."

>>Table 3 was referred

P6: "(Table 2 and Supplementary Table 1)" should be added in the end of the first sentence in section 3.2.

>>Table 1 and 2 were referred

Table 2, 3, and 4: "Species Name" in Table 2, "S/n" in Table 3, and "ID" in Table 4 seem to indicate same numbers. Use one item. The order of D. hansenii and D. prosopidis and the order of H. opuntiae and H. uvarum in Table 2 are same as in Table 3 but different in Table 4. Check carefully.

>> Corrected to Sample ID and Species Name.

Table 3: "Limon fruit" should be "Lemon fruit".

>> Changed to Lemon.

L1, 3, 4, 5: Lemon, Mango, and Guava should not be in italics.

>> Corrected

P10L4: Do not use capital letter for rhizosphere.

>> Corrected

P10L13: "(14)" should be "[14]".

>> I couldn’t get this word.

P10L41: Do not use capital letter for single cell protein.

>> Corrected.

P11L9: Candida, Pichia, Debaryomyces should be in italics.

>> Corrected